# *Carissa macrocarpa* Leaves Polar Fraction Ameliorates Doxorubicin-Induced Neurotoxicity in Rats via Downregulating the Oxidative Stress and Inflammatory Markers

**DOI:** 10.3390/ph14121305

**Published:** 2021-12-14

**Authors:** Mohamed A. A. Orabi, Heba M. A. Khalil, Mohamed E. Abouelela, Dalia Zaafar, Yasmine H. Ahmed, Reham A. Naggar, Hamad S. Alyami, El-Shaymaa Abdel-Sattar, Katsuyoshi Matsunami, Dalia I. Hamdan

**Affiliations:** 1Department of Pharmacognosy, College of Pharmacy, Najran University, Najran 55461, Saudi Arabia; maorabi@nu.edu.sa; 2Department of Veterinary Hygiene and Management, Faculty of Veterinary Medicine, Cairo University, Giza 12211, Egypt; 3Department of Pharmacognosy, Faculty of Pharmacy, Al-Azhar University, Assiut-Branch, Assiut 71524, Egypt; m_abouelela@azhar.edu.eg; 4Pharmacology and Toxicology Department, Faculty of Pharmacy, Modern University for Information and Technology, Cairo 11311, Egypt; dr.moda88@gmail.com; 5Cytology and Histology Department, Faculty of Veterinary Medicine, Cairo University, Giza 12211, Egypt; dr.yasmine.hamdy@gmail.com; 6Department of pharmacology and Toxicology, College of Pharmacy and Drug Manufacturing, Misr University of Science and Technology (MUST), 6th October, Giza 12566, Egypt; pookiemust@gmail.com; 7Department of Pharmaceutics, College of Pharmacy, Najran University, Najran 55461, Saudi Arabia; hsalmukalas@nu.edu.sa; 8Department of Medical Microbiology and Immunology, Faculty of Pharmacy, South Valley University, Qena 83523, Egypt; elshaymaa_a_m@svu.edu.eg; 9Department of Pharmacognosy, Graduate School of Biomedical and Health Sciences, Hiroshima University, 1-2-3 Kasumi, Minami-Ku, Hiroshima 734-8553, Japan; matunami@hiroshima-u.ac.jp; 10Department of Pharmacognosy and Natural Products, Faculty of Pharmacy, Menoufia University, Shibin Elkom 32511, Egypt

**Keywords:** *Carissa macrocarpa*, cytotoxicity, UPLC-ESI-MS/MS, doxorubicin, neurotoxicity, molecular docking

## Abstract

Chemotherapeutic-related toxicity exacerbates the increasing death rate among cancer patients, necessitating greater efforts to find a speedy solution. An in vivo assessment of the protective effect of the *C. macrocarpa* leaves polar fraction of hydromethanolic extract against doxorubicin (Dox)-induced neurotoxicity was performed. Intriguingly, this fraction ameliorated Dox-induced cognitive dysfunction; reduced serum ROS and brain TNF-α levels, upregulated the brain nerve growth factor (NGF) levels, markedly reduced caspase-3 immunoexpression, and restored the histological architecture of the brain hippocampus. The in vivo study results were corroborated with a UPLC-ESI-MS/MS profiling that revealed the presence of a high percentage of the plant polyphenolics. Molecular modeling of several identified molecules in this fraction demonstrated a strong binding affinity of flavan-3-ol derivatives with TACE enzymes, in agreement with the experimental in vivo neuroprotective activity. In conclusion, the *C. macrocarpa* leaves polar fraction possesses neuroprotective activity that could have a promising role in ameliorating chemotherapeutic-induced side effects.

## 1. Introduction

Cancer is a serious disease with a high percentage of deaths globally. According to the world health organization (WHO), it is considered the first or second cause of death in many countries, including those in the Middle East [1]. Cancer patients are subjected to many therapeutic modalities, including chemotherapeutic medications [2]. Despite the high efficacy of chemotherapeutic drugs, many patients suffered from chemotherapy-induced side effects [3]. Doxorubicin (Dox), an anthracycline drug, is used to treat many types of cancers, including breast cancer, esophageal and solid tumors [4]. It is worth noting that Dox cannot cross the blood-brain barrier. However, many preclinical studies reported neurotoxicity associated with its treatment [4,5,6].

The neurotoxicity associated with the chemotherapeutic medication is characterized by various symptoms, including cognitive dysfunction and depressive episodes [7]. Several attempts have been conducted to find and investigate new agents that can be complementary or adjuvant agents with chemotherapy to reduce their side effects. Natural plants have been taken great attention in recent decades being also useful in treating various types of cancer, owing to their polyphenols content [8].

The genus *Carissa* (Apocynaceae) comprises 36 species, including *Carissa macrocarpa* (Eckl.) A. DC. (*C. macrocarpa*) [9]. *C. macrocarpa* (commonly known as Natal plum) is an ornamental shrub that grows worldwide and is characterized by large, green, lush, and persistent leaves, white star-shaped flowers, and edible oval fruits [9]. Flavonoids, lignans, simple phenolic compounds, sterols, terpenes, esters, and fatty acids are the frequently reported phytochemical classes in *carissa* species [10].

Pharmacological studies on *Carissa* species have indicated significant anti-tumor, anticonvulsant, diuretic, vasorelaxant, anti-hyperlipidaemic, antioxidant, anti-inflammatory, antipyretic, analgesic, antiviral, antibacterial, antiplasmodial, anthelmintics, cardioprotective, hepatoprotective, antidiabetic, and antiemetic activities [11,12]. 

Polar extracts of different *carissa* species have revealed high efficacy against various human cancer cell lines, including human leukemia (HL-60), human cervical cancer (HeLa), human prostate cancer (PC-3), human ovarian carcinoma, and lung cancer cell lines [11,13,14]. The neuroprotective activities of a related *Carissa species* (*C. edulis* and *C*. *carandas*) were also reported [15,16,17]. Therefore, to exploit its role as a possible co-therapy with chemotherapeutic drugs, *C. macrocarpa* leaves polar fraction of the hydromethanolic extract was evaluated by an in vivo experiment for its neuroprotective effect against Dox-induced neurotoxicity. The UPLC-ESI-MS/MS metabolites profiling was also performed. Moreover, several polyphenolic compounds in the analyzed fraction, as well as their isomeric forms, were subjected to in silico molecular docking simulation for assessing their binding affinity to TNF-activating converting enzyme (TACE). 

## 2. Results

### 2.1. In Vivo Neuroprotective Activity of C. macrocarpa Leaves Polar Fraction against Dox-Induced Neurotoxicity

#### 2.1.1. Behavioural Parameters

Dox-exposed rats displayed reduced cognitive abilities, including short-term memory and long-term memory. In the Y-maze, Dox-exposed rats exhibited a significant decrease in the number of arm entries, and spontaneous alternation percentage (SAP%) compared to control rats. However, Dox+ 500 mg/kg C. *macrocarpa* leaves polar fraction (C.500) treated rats showed a notable increase in the number of arm entries compared to Dox-exposed rats. Meanwhile, there was no significant difference in the number of arm entries between Dox+ C.100, Dox+ C.300 treated rats, and Dox-exposed rats. Concerning SAP%, Dox+ C.300 and Dox+ C.500 treated rats displayed a marked increase in the SAP% compared to Dox-exposed rats. However, there was no significant difference between Dox+ C.100 treated rats and Dox-exposed rats (Figure 1a,b). In addition, Dox-exposed rats displayed a marked decrease in the duration of exploration and the preference for a new object compared to control rats. Meanwhile, Dox+ C.100, Dox+ C.300, and Dox+ C.500 treated rats showed a marked elevation in the exploration duration compared to Dox-exposed rats in a dose-dependent manner. Concerning the preference for the new object, Dox+ C.500 treated rats showed a marked increase in the preference for the new object compared to Dox-exposed rats. Meanwhile, there was no significant difference in the preference for the new object between Dox+ C.100, Dox+ C.300 treated rats, and Dox-exposed rats (Figure 1c,d).

#### 2.1.2. Biochemical Parameters

Dox-exposed rats displayed marked elevated serum levels of ROS and TNF-α. However, Dox+ C.100, Dox+ C.300, and Dox+ C.500 treated rats exhibited substantially reduced ROS levels in a dose-dependent manner, demonstrating the antioxidant and anti-inflammatory effect gained upon adding the C. *macrocarpa* leaves polar fraction (Figure 2a,b). On the other hand, brain levels of NGF were severely down-regulated in Dox-exposed rats. However, in Dox+ C.100, Dox+ C.300, and Dox+ C.500 treated rats, NGF levels elevated significantly in a dose-dependent manner (Figure 2c).

#### 2.1.3. Histopathological and Immunohistochemical Investigations

##### Histological Examination

H and E-stained hippocampus sections of control rats showed the normal histological structure of the three layers; molecular, pyramidal, and polymorphic cell layers. The molecular layer contained scattered neuronal cells, neuroglia, and blood capillaries. The pyramidal cell layer consisted mainly of pyramidal neurons with triangular-shaped cell bodies and a vesicular spherical nucleus with a prominent nucleolus. The polymorphic cell layer is made up of pyramidal cells, neuroglia cells, and blood capillaries on the neuropil background (Figure 3a).

On the other hand, the hippocampus sections obtained from Dox-exposed rats revealed several histopathological alterations such as vacuolation of neuropil, pyknotic, and hyperchromatic neuroglia cells. Some pyramidal cells had neurofibrillary tangles and were surrounded by pericellular space. Additionally, some pyramidal cells that appeared severely damaged and degenerated showed dystrophic changes; they were shrunken with hyperchromatic and pyknotic nuclei hyperchromatic. Wide perivascular spaces with congested blood capillaries were observed (Figure 3b).

Meanwhile, the hippocampus sections of rats treated with Dox+ C.100, Dox+ C.300, and Dox+ C.500 showed partial recovery compared to Dox-exposed rats, evidenced by decreased neuropil vacuolation, perineuronal, and perivascular spaces. Pyramidal cells restored their triangular shape with a vesicular spherical nucleus, but few pyramidal cells degenerated with the pericellular space (Figure 3c–e).

##### Immunohistochemical Examination for Caspase-3

Immunohistochemistry of hippocampus samples from control rats stained by caspase-3 revealed negligible immunoreactivity. Meanwhile, an intense positive immunoreaction associated with a significant increase in the percentage of area covered by caspase-3 positive immunoreactive cells within the hippocampus was noticed in Dox-exposed rats compared to control rats. A marked reduction of caspase-3 immunoexpression and covered area % was observed in Dox+ C.100 treated rats compared to Dox-exposed rats. In addition, Dox+ C.300 and Dox+ C.500 treated rats showed negligible caspase-3 immunoexpression and covered area % compared to Dox-exposed rats (Figure 4).

### 2.2. UPLC-ESI-MS-MS Metabolites Characterization

Herein, 43 metabolites were tentatively identified in the polar fraction of C. macrocarpa leaves using UPLC-ESI-MS/MS in negative and positive ionization modes (Appendix A). The compounds were identified based on their MS and MS^2^ fragmentation data and the comparison with the literature. The identified metabolites are incontrovertibly classified into several groups: organic acid derivatives, flavonoid aglycones and glycosides, flavan-3-ol derivatives, sterols and triterpenes, and miscellaneous. The compounds are summarized and ordered according to their retention time (R_t_) in Table 1. Although the majority of these identified compound was reported as the phytomolecules of C. macrocarpa by actual isolation and LCMS/MS investigations [9,11,18,19], their precise identity herein is based on the permitted % mass error is a limitation of this study.

### 2.3. Molecular Docking of Polyphenolic Compounds against TACE

The in silico molecular docking of some phytoconstituents identified by UPLC-ESI- MS/MS in C. macrocarpa polar fraction were performed to evaluate their potential against Dox-induced neurotoxicity. The analysis of molecular simulation results for TACE revealed that procyanidin B6, epicatechin 3-O-β-D-glucopyranoside, procyanidin B5, and hyperoside were the top-scoring compounds with pose scores −22.0937, −21.9152, −20.6389, −19.9566, and −19.9102 (kcal/mol), respectively (Appendix A). The procyanidin B6 showed the highest affinity score with −21.9152 (RMSD = 1.25). The interaction of the compound with the target enzyme (Figure 5) showed the presence of hydrogen bond formation with Arg 357, Pro 437, and Glu 406 amino acid residues, as well as ionic interaction with HIS 405, HIS 409, HIS 415, and zinc ion. It also showed π-H and π-π interactions with Thr 347 and His 405, respectively.

## 3. Discussion

The increasing deaths among cancer patients provoked by chemotherapeutics-related side effects necessitate increased efforts to find an urgent solution. Plant polyphenols are the most extensively researched anti-cancer candidate. Their popularity is increasing as a co-therapy with standard cancer chemotherapeutics to improve their efficacy and reduce their side effect [8,32,33].

The hydro-ethanolic extract from *C. macrocarpa* fruits and leaves have been shown to inhibit the growth of HeLa, NCI-H460 (non-small cell lung carcinoma), and MCF-7 cancer cell lines [9]. Besides, polar fraction of morphological parts of related *Carissa* species (*C. edulis* and *C. carandas*) have demonstrated neuroprotection against various induced neurotoxicity. Taken together, the MeOH fraction, which made up the majority (84.3%) of the total hydromethanolic extract of *C. macrocarpa* leaves, was tested against Dox-induced neurotoxicity in order to investigate its potential as a feasible co-therapy with chemotherapeutics. In our study, Dox administration caused marked cognitive impairments in both short and long term memories visualized by a reduction in the number of arm entries and spontaneous alternation percentage, which is considered a common behaviour in rodents to alternate between three different arms in the Y-maze, in consistence with literature [34,35]. In addition, we used a novel object recognition test to detect the discrimination criteria in the rodents. Dox-exposed rats displayed a reduction in the total exploratory time as well as the preference for the novel object. These findings were also in agreement with those reported in the references [35,36]. Conversely, in a dose-dependent way, *C. macrocarpa* in three distinct doses, 100, 300, and 500 mg/kg, was able to alleviate short and long-term memory deficits (Figure 1).

Dox-induced cognitive impairments were confirmed by the histopathological examination of the hippocampus, which was severely damaged and degenerated. Pyramidal cells showed dystrophic changes that were shrunken with pyknotic nuclei and surrounded by pericellular space. These findings corroborated those of the study by Leung et al., 2020 [37], where degeneration of cellular constituents of hippocampus was indicated by shrunken stained neurons and enlarged interstitial spaces of Dox-exposed rats. According to Tangpong et al., 2006 [38], Dox exposure might cause neuronal damage by increasing oxidative stress and plasma levels of TNF-α, which can pass the blood-brain barrier and trigger glial cells to produce more TNF-α in the brain, resulting in neuroinflammation and oxidative stress, which can lead to brain damage [38]. Additionally, neuropil vacuolation and wide perivascular spaces were observed in the cerebral cortex of rats injected intraperitoneally with 4 mg/kg Dox in the present work that comes in line with previously reported observations [39]. The vacuolation of the neuropils may be attributed to the shrinkage of cells and withdrawal of their process’s secondary to cytoskeletal affection, thereby leaving pericellular spaces [40]. However, the hippocampus sections of rats treated with *C. macrocarpa* 100, 300, and 500 mg/kg) combined with Dox showed partial recovery compared to Dox-exposed rats as evidenced by decreased neuropil vacuolation, perineuronal and perivascular spaces. Pyramidal cells restored their triangular shape with a vesicular spherical nucleus, but few pyramidal cells degenerated with the pericellular space (Figure 3). This regeneration may attribute to the high antioxidant and anti-inflammatory activities of the *C. macrocarpa* leaves [9,19].

Upregulation of ROS is hypothesized to be one of the Dox neurotoxicity routes since neurons are vulnerable to oxidative stress [41]. Our findings indicate that Dox-exposed rats developed oxidative brain damage, as characterized by the elevation of ROS serum levels. Several studies proved the link between oxidative stress and inflammation. The current study showed that the oxidative stress and inflammatory state caused by administration of Dox were successfully down-regulated upon adding different doses (100, 300, and 500 mg/kg) of *C. macrocarpa* in a dose-dependent manner, and the lowest oxidative stress and pro-inflammatory state were shown in the group administering the highest dose of *C. macrocarpa* (Dox+ C.500 mg/kg (Figure 2a,c). Reducing levels of ROS and TNF-α are closely associated with reducing neurotoxicity, as mentioned in several recent studies [37,42]. Furthermore, the produced TNF-α by Dox can cross the blood-brain barrier and activate the microglia in the brain to release more TNF-α that elevates the production of nitric oxide by glia, leading to nitrosative and oxidative stress, mitochondrial dysfunction, endoplasmic reticulum stress, and neuronal apoptosis [5,43,44]. The neuronal apoptosis was evidenced by the significant increase in the caspase-3 immunoreactivity in the Dox-exposed group compared to the control group. However, there was a significant reduction of caspase-3 immunoreaction in the hippocampus section of rats treated by *C. macrocarpa* with three different doses (100, 300, and 500 mg/kg) and Dox (Figure 4). This may be due to the antioxidant and anti-inflammatory activities of the *C. macrocarpa* leaves, which suppress Dox toxicity by scavenging reactive oxygen species and decreasing TNF-α production resulted in reducing the apoptotic process.

Several studies highlighted the crucial role of NGF in neurogenesis, cognition, and brain repair, indicating the importance of maintaining NGF levels to prevent the occurrence of neurotoxicity [45,46,47]. In the current study, the lowest levels of NGF obtained by administering Dox alone were successfully elevated after adding *C. macrocarpa* combined with Dox. In addition, the highest dose of *C. macrocarpa* (500 mg/kg) returned brain NGF levels to normal levels (Figure 2b).

The presence of polyphenolic metabolites [flavonoids, flavan-3-ols (catechins and epicatechins), phenolic acids, and their derivatives] as detected by UPLC-ESIMS/MS and confirmed in a previous study [19] will consequently leads to high antioxidant and anti-inflammatory properties, which may contribute to their neuroprotective properties [4]. It is worth mentioning that flavan-3-ol polyphenols, a major component in our active polar fraction, have been shown to activate a variety of cellular processes that contribute to their neuroprotective properties, such as iron chelation, survival gene and cell signaling pathway activation, and mitochondrial activity control [48,49,50,51]. As a result, these polyphenolics are anticipated to be among the most important metabolites involved in dox-induced toxicity neuroprotection. The neuroprotective activity of the *C. macrocarpa* leaves polar fraction may be due to the presence of the polyphenolic metabolites [4] (flavonoids, flavan-3-ols, phenolic acids, and their derivatives) detected by the herein UPLC-ESIMS/MS and confirmed in previous researches that consequently leads to high antioxidant and anti-inflammatory properties [48,49,50,51]. Although other *Carissa* species have been shown to have neuroprotective properties, this is the first report on *C. macrocarpa* neuroprotection. [15,17].

It should be noted that TNF-α serum level, a pro-inflammatory mediator that contributed to neurotoxicity, depends on the rate of its synthesis as well as on its shedding from the cell surface (a mechanism mainly regulated by TACE). Molecular docking is frequently used to rationalize the inhibitory potential of compounds against selected target proteins and to characterize their behaviour in the binding site to predict affinity and activity [52,53]. Altogether, to corroborate the reported in vivo activity, in silico inhibitory activity experiment of variously identified phytoconstituents against TACE was performed. Procyanidins are among the highest-scoring compounds as inhibitors for TACE, in which the phenolic hydroxyl groups and aromatic rings impart in the interaction with the active site residues. Although few compounds were studied, these results put focus on structural characteristics that best contribute to the activity of such drugs as TACE inhibitors. The results of simulation studies are consistent with the reported activity of procyanidins, where a procyanidin enriched extract revealed a reduction in TNF-α levels and suppressed the LPS-stimulated inflammatory response in the RAW264.7 cells [54]. Further, this will add to our understanding of the flavonoid structural requirements and the pharmacophores concerned with the receptor-protein complex. Our results, in agreement with those previously detected for the same and/or analogous compounds by experimental and molecular docking studies [4,55], suggest that polyphenolic compounds of the *C. macrocarpa* leaves polar fraction with elevated pose score could be used as a scaffold for developing new inhibitors for enzymes related to neurotoxicity (TACE).

## 4. Materials and Methods

### 4.1. Plant Material

*C. macrocarpa* leaves (Eckl.) A. DC were collected in October 2019 from a mature tree in the flowering stage growing in the front of Faculty of Engineer, Najran, Kingdom of Saudi Arabia (https://maps.app.goo.gl/4vnxmEuyqaWGuktc6, accessed on 9 August 2021). A voucher specimen (CM 1019) was deposited in the Pharmacognosy Department, College of Pharmacy, Najran University.

### 4.2. Extraction and Fractionation

Air-dried powdered leaves of *C. macrocarpa* (250 g) were homogenized in MeOH–H_2_O (8:2, *v*/*v*, 4 × 1.5 L) at room temperature. The hydroalcoholic extract was dried at 40 °C under reduced pressure. The majority (56 g) of the obtained total dry methanolic extract (66.6 g) was fractionated by flash chromatography column packed with silica gel (70–230 mesh, Merck, Darmstadt, Germany) (15 × 7 cm, i.d.), and eluted with *n*-hexane, EtOAc, *n*–butanol, and MeOH (3 L each), successively. Vacuum drying of the different eluates gave the corresponding dry fractions (0.2, 5.52, 3.63, and 47.2 g, respectively). The major MeOH fraction (47.2) used in the current study was kept in clean sample vials for biological studies and phytochemical profiling.

### 4.3. In Vivo Evaluation of the C. macrocarpa Leaves Polar Fraction against Dox-Induced Neurotoxicity

#### 4.3.1. Animals

Wistar male rats (120–160) were obtained from the breeding unit of Veterinary Hygiene and Management, Faculty of Veterinary Medicine, Cairo University (Giza, Egypt. Rats were maintained in a 12-h day/night cycle, room temperature (25 ± 2 °C), and humidity (50%) with free access to a commercial balanced diet and water. They were acclimatized to the breeding facility environment before proceeding with the experimental study.

#### 4.3.2. Acute Toxicity Study

An acute oral toxicity study was conducted using the limit test procedure according to the Organization for Economic Co-operation and Development (OECD) guidelines for evaluating chemicals [56]. According to the procedure, Wistar rats were randomly divided into six groups of six animals each. Different doses (100, 300, 500, 1000, 2000, and 4000 mg/kg) of *C. macrocarpa* MeOH fraction of the leaves were administered orally using oral gavage. Then, the rats were observed for 72 h in accordance with the reported procedure [57] for any behavioral changes, signs of toxicity, and death. Since there were no mortalities or any physical or behavioural abnormalities, 100, 300, and 500 mg/kg body weight of the *C. macrocarpa* leaves polar fraction were selected for the biological evaluation.

#### 4.3.3. Experimental Design

Rats were randomly assigned to one of five groups (*n* = 7 per group). Group I (control) received orally administered 2 mL/kg per day of water for 4 weeks; Group II (Dox) received doxorubicin (Mylan pharmaceutical, Egypt, 2.5 mg/kg/week/i.p) for 4 weeks; Group III (Dox+ C.100) received orally administered 100 mg/kg per day of *C. macrocarpa* and Dox (2.5 mg/kg/week/i.p) for 4 weeks; Group IV (Dox+ C.300) and group V (Dox+ C. 500) received orally administered 300, and 500 mg/kg per day of *C. macrocarpa*, respectively and Dox (2.5 mg/kg/week/i.p) for 4 weeks. The experimental protocol was conducted according to the literature [58,59].

#### 4.3.4. Behavioural Tasks

Twenty-four hours after the last dose of treatment, rats were submitted to cognitive assessment where Y-maze and novel object recognition test were performed to measure the short term working memory and long term memory, respectively, according to reported methods [60,61]. The rats were observed for 5 min, and the recorded parameters for the Y-maze were the number of arm entries and SAP%. While for the novel object recognition test, the total exploration duration and preference for the novel object were evaluated.

#### 4.3.5. Euthanasia and Sample Collection

After the last behavioural test, blood was collected from the inner canthus of the rat’s eye for serum separation and preserved at −20 °C for the subsequent biochemical analysis. Then the rats were euthanized by cervical decapitation, and the brains were excised gently. Half of the brains were fixed in 10% neutral buffer formalin for 48 h and prepared for histopathological and Immunohistochemical studies. While the other halves were preserved at −80 °C for the subsequent assays.

#### 4.3.6. Biochemical Parameters

##### Examination of Serum Levels of Reactive Oxygen Species (ROS)

ROS serum levels were determined using an ELISA kit (Catalog No. LS-F9759) obtained from Lifespan bioscience (Seattle, WA 98121, USA) according to the manufacturer’s instructions.

##### Examination of TNF-α Pro-Inflammatory Mediator in the Brain Tissue

The brain levels of the pro-inflammatory cytokine TNF-α were measured using a commercial ELISA kit (Catalog No.: abx050220) obtained from Abbexa Merck Millipore (Cambridge, England) according to the manufacturer’s instructions.

##### Estimation of Nerve Growth Factor (NGF) Levels in the Brain Tissue

Brain levels of NGF were estimated using an ELISA kit (Catalog no. SEA105Ra) obtained from Cloud-clone corp. (Houston, TX, USA) according to the manufacturer’s instructions.

#### 4.3.7. Histopathological and Immunohistochemical Investigations

##### Histological Examination

The fixed samples were dehydrated in ascending grades of ethyl alcohol, cleared in xylene, and embedded in paraffin wax. Sections of 3–4 μm in thickness were obtained by rotatory microtome, deparaffinized, and stained with hematoxylin and eosin (H&E) stain for examination under a light microscope [62].

##### Immunohistochemical Examination for Caspase-3

The avidin-biotin-peroxidase technique was used for the detection of activated caspase-3 as an apoptotic marker [63]. Brown cytoplasmic or nuclear staining is considered a positive reaction. After deparaffinization and rehydration of the cerebellar sections, antigen retrieval was achieved by boiling sections in citrate buffer in a microwave. Endogenous peroxidase was blocked using H_2_O_2_. After blocking non-specific background with 10% serum-tris buffer for 20 min at room temperature, the sections were then incubated with the primary antibody anti-caspase-3 rabbit polyclonal antibody (Catalogue No. RB-1197, Thermo-Fisher Scientific) diluted 1/100 at room temperature for 120 min. The slides were subsequently incubated with a biotinylated polyvalent secondary antibody and then incubated with avidin-biotin-peroxidase complex solution (LSAB2 Kit, Dako). The reaction was visualized by adding 3, 3′-diaminobenzidine tetrachloride to the sections. Hematoxylin was used to counterstain the sections. Slides stained with secondary antibody IgG only were used as negative controls. Specimens from palatine tonsils were used as positive controls.

Immunohistochemically stained sections were examined using Leica Quin 500 analyzer computer system (Leica Microsystems, Switzerland) in the Faculty of Dentistry, Cairo University. The image analyzer was calibrated automatically to convert the measurement units (pixels) produced by the image analyzer program into actual micrometer units. Immunohistochemical reactions were measured as a percentage of area in a standard measuring frame in 5 fields of different slides in each group using magnification (×400) by light microscopy transferred to the monitor screen. The areas showing the positive Immunohistochemical reaction were chosen for evaluation, regardless of the intensity of the staining. These areas were masked by a blue binary color to be measured by the computer system. Mean values and standard deviation were obtained for each specimen and statistically analyzed.

### 4.4. UPLC-ESI-MS/MS Analysis of C. macrocarpa Leaves Polar Fraction

The MeOH fraction (100 μg/mL) solutions were prepared using high-performance liquid chromatography (HPLC) analytical grade solvent of MeOH followed by filtration using a 0.2 μm membrane disc filter and subjected to UPLC-ESI-MS/MS analysis according to the reported procedure [64].

### 4.5. Molecular Docking

Docking analysis was conducted using Molecular Operating Environment software (MOE 2014.0901). The crystal structures of TNCE (PDB ID: 2FV5) [65] were obtained from the Protein Data Bank (https:/www.rscb.org/pdb, accessed on 9 August 2021) [66]. The proteins receptor structures were prepared and optimized for docking by the MOE Ligx option. The receptor’s active sites for ligand binding were determined based on amino acid residues interacted with the complexed ligand for each protein. Unessential residues and water molecules were removed. The compounds summarized in Appendix A and illustrated in (Appendix A) were imported to MOE and subjected to energy minimization using MMFF94x force field, and a virtual ligand database was constructed. The molecular docking simulation was performed through flexible ligand-fixed receptor docking using Triangle Matcher placement, forcefield refinement with London dG as scoring, and rescoring algorithm. The docking score, root mean square deviation (RSMD), 2D, and 3D interactions were recorded.

## 5. Conclusions

The polar fraction obtained from *C. macrocarpa* leaves exhibited a potential neuroprotective activity against Dox-induced neurotoxicity as evidenced by reducing the cognitive deficits, ROS serum levels, and TNF-α brain levels, as well as elevating NGF brain levels. Moreover, the reduction in the caspase-3 immunoreactivity associated with restoring the normal brain histological architecture was demonstrated. The results may be attributed to polyphenolic identified compounds (epicatechin/catechin glycosides, procyanidins, flavonoids, and phenolic acid derivatives) that could be a therapeutic alternative to treat chemotherapeutic drug side effects. These results were supported by a molecular docking study. Further studies to screen ligands that support the clinical phase (pharmacokinetics) are warranted. Additionally, dose determination, safety, and efficacy studies will be performed as an initial step for the clinical trial of *C.macrocarpa* leaves.

## Figures and Tables

**Figure 1 pharmaceuticals-14-01305-f001:**
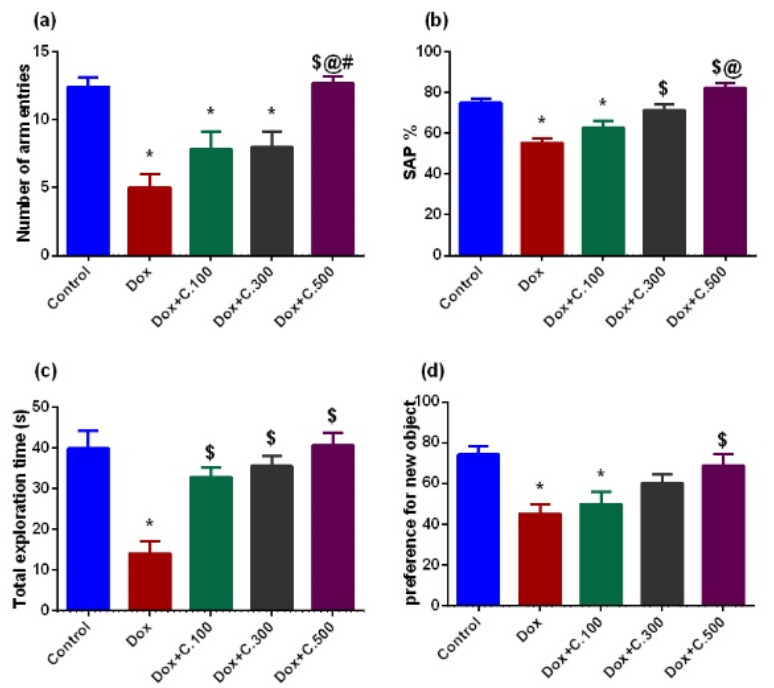
Effect of Dox with or without different doses of *C. macrocarpa* leaves polar fraction on the cognitive functions of rats. (**a**) Y-maze: Number of arm entries, (**b**) Y-maze: Spontaneous alternation percentage, (**c**) Novel object recognition test: Total exploration time, and (**d**) Novel object recognition test: Preference for new object. Data are expressed as mean ± SEM, one-way ANOVA followed by post-hoc test Tukey test for seven rats in each group. * Significant from control group, $ Significant from Dox group, @ Significant from Dox+ C.100 group, and # Significant from Dox+ C.300 group, *p <* 0.05.

**Figure 2 pharmaceuticals-14-01305-f002:**
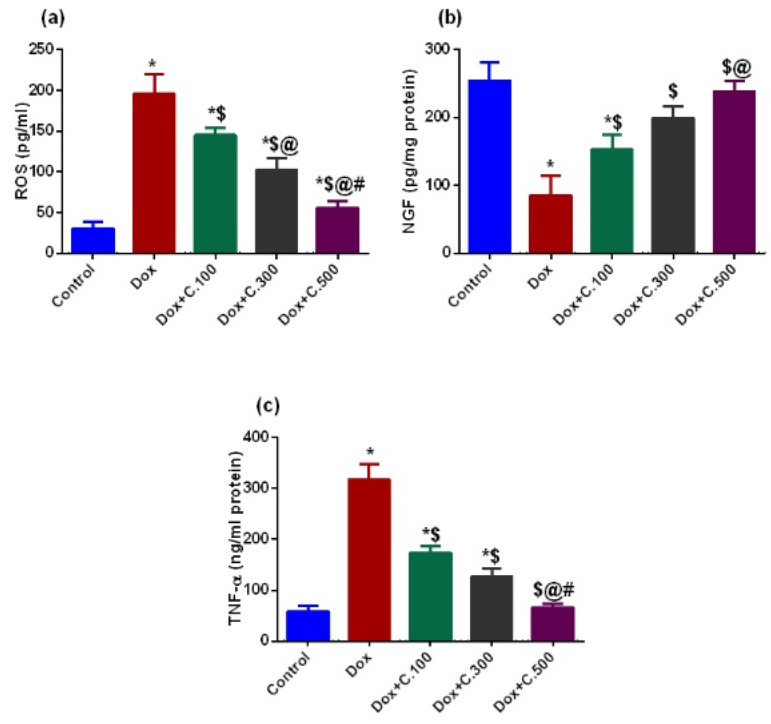
Effect of Dox with or without different doses of *C. macrocarpa* leaves polar fraction on the brain levels of ROS, NGF, and TNF-α. (**a**) Reactive oxygen species, (**b**) Nerve growth factor, and (**c**) Tumor necrosis factor-alpha. Data are expressed as mean ± SEM, one-way ANOVA followed by post hoc Tukey test for seven rats in each group. * Significant from control group, $ Significant from Dox group, @ Significant from Dox+ C.100 group, and # Significant from Dox+ C.300 group, *p <* 0.05.

**Figure 3 pharmaceuticals-14-01305-f003:**
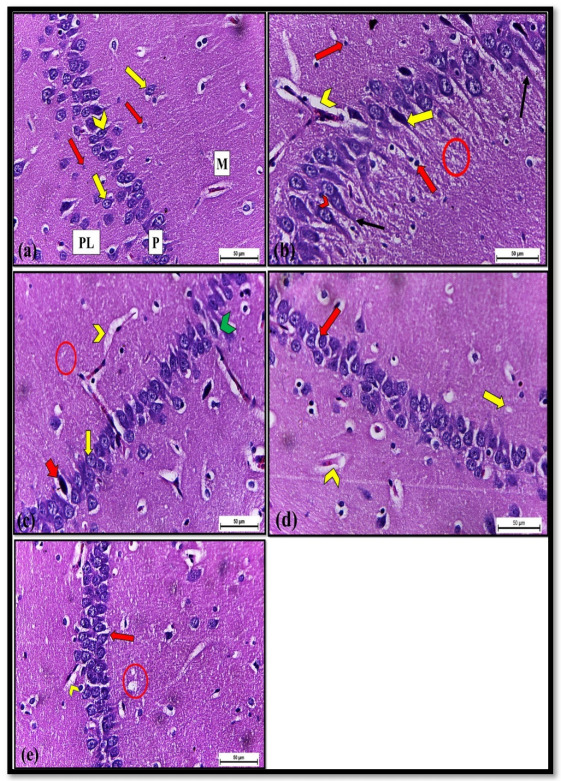
Brain tissue sections of albino rats from the hippocampal region H and E ×400. (**a**) Control rats (Group Ⅰ) showed normal molecular layer (M), pyramidal cell layer (P), polymorphic cell layer (PL). The molecular layer contained neuronal cells (yellow arrow) and neuroglia cells (red arrow). The pyramidal layer consisted of triangular cells (chevron), and the polymorphic cell layer contained neuronal cells (yellow arrow) and neuroglia (red arrow). (**b**) Dox-exposed rats (Group Ⅱ) revealed histopathological changes such as neuropil vacuolation (circle), pyknotic neuroglia cells (red arrow), degenerated, shrunken and hyperchromatic pyramidal cell (yellow arrow). A wide perivascular space with congested blood capillary (yellow chevron) was observed. Additionally, some pyramidal cells with neurofibrillary tangles (line arrow) and pericellular space (red chevron). (**c**) Dox+ C100 treated rats (Group Ⅲ) revealed partial recovery compared to group Ⅱ in the form of diminishing of neuropil vacuolation (circle), perivascular space (yellow chevron), and perineuronal space (green chevron). Pyramidal cells restored their triangular shape with a vesicular nucleus (yellow arrow), but few pyramidal cells appeared degenerated (red arrow). (**d**) Dox+ C300 (Group Ⅳ) and (**e**) Dox+ C500 (Group Ⅴ) treated rats showed regeneration signs same to group Ⅲ in the form of nearly normal pyramidal cells (red arrow), few neuropil vacuolations (yellow arrow and circle), and reduced perivascular space (yellow chevron).

**Figure 4 pharmaceuticals-14-01305-f004:**
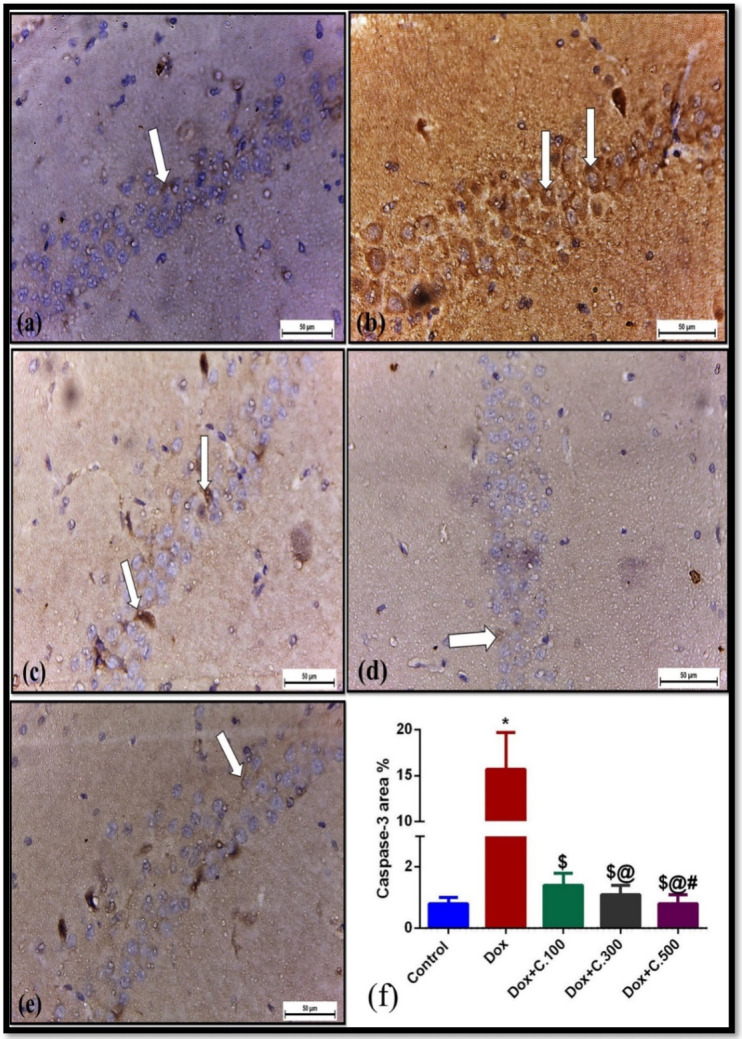
Immunohistochemically caspase-3-stained hippocampus sections (×400). (**a**) Control rats (Group Ⅰ) showing negligible immunoreactivity to caspase-3 (arrow). (**b**) Dox-exposed group (Group Ⅱ) showing strong immunoexpression to caspase-3 (arrow). (**c**) Mild immunoreactivity to caspase-3 (arrow) in Dox+ C.100 administered rats (Group Ⅲ). (**d**) Dox+ C.300 (Group Ⅳ) and (**e**) Dox+ C.500 (Group Ⅴ) treated rats showing negligible immune reaction (arrow). (**f**) A bar graph showing a significant increase in the area% covered by Caspase3-positive immunoreactive cells within the hippocampus in Dox-exposed albino rats (Group Ⅱ) compared to the control rats (Group Ⅰ) and a significant decrease in Dox + *C. macrocarpa* treated rats (Groups Ⅲ, Ⅳ, and Ⅴ) compared to Dox-exposed rats. Data are presented as mean values ± SEM, one-way ANOVA followed by post hoc test Tukey test for seven rats in each group. * Significant from control group, $ Significant from Dox group, @ Significant from Dox+ C.100 group, and # Significant from Dox+ C.300 group, *p <* 0.05.

**Figure 5 pharmaceuticals-14-01305-f005:**
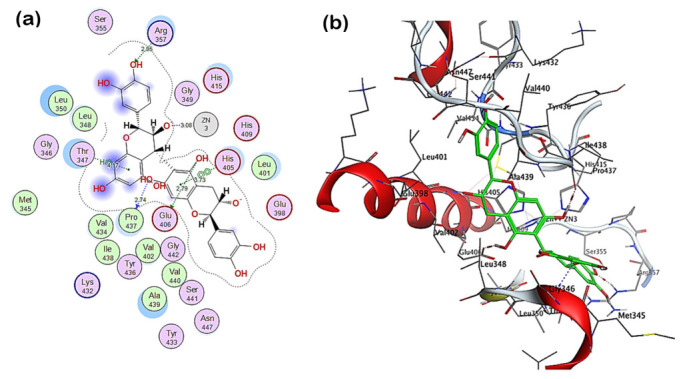
2D (**a**) and 3D (**b**) interactions complex of procyanidin B6 with TACE (PDB ID: 2FV5).

**Table 1 pharmaceuticals-14-01305-t001:** Secondary metabolites identified in the *C. macrocarpa* leaves polar fraction using UPLC-ESI-MS-MS.

Peak No.	R_t_	[M−H]^−^	[M+H]^+^	MS^2^	Tentative Identification	References
1	0.74	377.16	379.25	333, 271, 257, 163, 119	Carinol	[20]
2	1.87	353.16	355.13	191,179,161	3-O-Caffeolyquinic acid	[9]
3	2.03	353.17	-	191,173,161	4-O-Caffeolyquinic acid	
4	2.13	325.14	-	187, 163, 145	Coumaroyl-*β*-glucose	[21]
5	2.161	353.20	-	191,179,161	5-O-Caffeolyquinic acid	[22]
6	2.24	577.27	-	425, 289	Type B (epi)catechin dimer	[9]
7	2.45	865.50	-	451, 425, 407, 289	Type B (epi)catechin trimer	
8	2.85	319.17	-	301, 275, 257, 231, 203,163, 119	5-O-*p*-Coumaroylshikimic acid	[23]
9	2.87	-	343.18	326, 311, 285	Caffeic acid 3-glucoside	[24]
10	2.87	451.30	-	408, 393, 351, 337, 301, 273, 245	Catechin-3-O-glucoside	[25]
11	2.87	319.17	-	275, 257, 199, 163, 119	4-O-*p*-Coumaroylshikimic acid	[23]
12	2.87	451.31	-	391, 343, 301, 287, 273, 247	Epicatechin-3-O-glucoside	[25]
13	3.00	289.09	-	245, 205, 203, 187, 179, 161	(epi) Catechin	[26]
14	5.05	755.44	-	593, 285	Kaempferol-7-O-hexoside-3-O-rutinoside	[9]
15	5.12	755.52	-	609, 301	Quercetin-7-O-deoxyhexoside-3-O-deoxyhexosyl-hexoside	
16	5.14	451.37	-	391, 343, 301, 287, 273, 247	Epicatechin-3-O-glucoside isomer	[25]
17	5.52	739.40	-	593, 285	Kaempferol-7-O -deoxyhexoside-3-O -deoxyhexosyl-hexoside isomer 1	[9]
18	5.59	739.42	-	593, 285	Kaempferol-7-O-deoxyhexoside-3-O-deoxyhexosyl-hexoside isomer 2	
19	5.59	609.27	-	301	Quercetin-3-O-deoxyhexosyl-hexoside isomer 1	
20	5.75	609.33	611.29	465, 303	Quercetin-3-O-deoxyhexosyl-hexoside isomer 2	
21	5.80	577.29	-	425, 289	Type B (epi)catechin dimer	
22	5.90	449.17	-	317, 316	Myricetin-3-O-xyloside	[23]
23	6.04	-	302.89	275, 257, 229, 215, 153	Quercetin	
24	6.13	593.33	-	557, 467, 441, 425, 407, 289	(epi) Gallocatechin-(epi)catechin	[27]
25	6.58	515.30	-	353, 179	Dicaffeoylquinic acid	[22]
26	6.80	136.94	-	109, 93	Hydroxy benzoic acid	[28]
27	7.01	593.33	-	557, 467, 441, 425, 407, 289	(epi) gallocatechin-(epi)catechin	[27]
28	8.12	196.93	-	120, 104, 93, 87	Syringic acid	[28]
29	9.14	573.70	-	397, 223, 173	Feruloyl-O-sinapoylquinic acid	[22]
30	9.94	939.06	-	778, 735, 732, 717, 571	Diacetoxy-5-methoxyphenyl) acroyl-O-*p*-coumaroyl-O-caffeoylquinic acid derivative	[23]
31	13.26	577.48	-	425, 289	(epi) Catechin dimer	[9]
32	13.30	543.56	-	353, 173	Dimethoxycinnamoyl-O-caffeoylquinic acid	[22]
33	15.66	543.33	-	353, 173	Dimethoxycinnamoyl-O-caffeoylquinic acid isomer	
34	16.52	352.99	-	179,161	3-O-Caffeoylshikimic acid	
35	16.59	-	383.25	369, 351, 195	Dimethoxycinnamoylquinic acid	
36	18.58	455.47	-	439, 419, 411, 410, 407, 397	Ursolic acid	[11,29]
37	19.06	455.46	-	439, 419, 411, 410, 407, 397	Carissic acid (isomer of ursolic acid)	
38	19.16	455.50	457.43	439, 419, 411, 410, 407, 397	Oleanolic acid	
39	25.30	-	413.31	395, 256, 214	Stigmasterol	[30]
40	27.02	-	465.45	301, 300, 257, 255, 229, 179. 151	Hyperoside	[31]
41	27.21	-	465.42	301, 300, 257, 255, 229, 179. 151	Isoquercetin	
42	27.31	621.68	-	501	2(R)-26-([(2E)-3-(4-hydroxy-3-methoxyphenyl)-1-oxo-2- propen-1-yl]oxy)-2,3-dihydroxypropyl ester	[10]
43	31.25	429. 31	430.92	205, 191, 177, 149, 121	α-Tocopherol	

## Data Availability

The data is contained within the article and Appendix A.

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
