# Peer review of "Carissa macrocarpa Leaves Polar Fraction Ameliorates Doxorubicin-Induced Neurotoxicity in Rats via Downregulating the Oxidative Stress and Inflammatory Markers"

_pharmaceuticals, 2021, doi:10.3390/ph14121305_

Round 1

Reviewer 1 Report

The Authors present the results from their experiments on protective effects of the compounds from Carissa macrocarpa leaves in Doxorubicin treated animals.

The in vivo and ex vivo results are quite interesting and well discussed. The protective effects of the applied treatment against Doxorubicin-induced neurotoxicity seems to be proven.

However, I see some problems in the parts of the manuscript concerning MS/MS profiling and molecular docking. To me, the discussion of these results should be extended. The present discussion is rather scarce:

„The neuroprotective activity of the C. macrocarpa leaves polar fraction may be due to the presence of the polyphenolic metabolites [4] (flavonoids, flavan-3-ols, phenolic acids, and their derivatives) detected by the herein UPLC-ESIMS/MS and confirmed in previous researches that consequently leads to high antioxidant and anti-inflammatory properties [48–51].

Besides, some sentences are unclear to me, e.g.:

„The results of the in-vivo study were corroborated with a UPLC-ESI-MS/MS profiling that revealed the accumulation of a high percentage of the plant polyphenolics.” What does it mean „accumulation”?

„The crystal structures of TNCE (PDB ID: 2FV5)”…. What does it mean „TNCE”?

„The procyanidin B6 showed the highest affinity score with −21.9152 (RMSD = 1.25).” The interaction of the compound with the target enzyme (Figure 5) showed the presence of hydrogen bond formation with Arg 357, Pro 437, and ….”

and

„Figure 5. 2D (A) and 3D (B) interactions complex of procyanidin C2 with TACE (PDB 212 ID: 2FV5)”.

I am not sure which compound is concerned: Procyanidin B6 or C2?

Author Response

Response to Reviewer 1

First of all the authors would like to thank the reviewer for their excellent comments that increased the significance and quality of our manuscript. We addressed all comments as outlined below and all the required adjustments were considered in the manuscript.

Point 1: However, I see some problems in the parts of the manuscript concerning MS/MS profiling and molecular docking. To me, the discussion of these results should be extended. The present discussion is rather scarce:

“The neuroprotective activity of the C. macrocarpa leaves polar fraction may be due to the presence of the polyphenolic metabolites [4] (flavonoids, flavan-3-ols, phenolic acids, and their derivatives) detected by the herein UPLC-ESIMS/MS and confirmed in previous researches that consequently leads to high antioxidant and anti-inflammatory properties [48–51].

Response 1: We thank the reviewer for the comment.

We have added a detailed discussion concerning the relation between the MS/MS characterized compounds and their possible contribution to the neuroprotective activity as follows:

“The presence of polyphenolic metabolites [flavonoids, flavan-3-ols (catechins and epicatechins), phenolic acids, and their derivatives] as detected by UPLC-ESIMS/MS and confirmed in previous study [19] will consequently leads to high antioxidant and anti-inflammatory properties, which may contribute to their neuroprotective properties [4].

It is worth mentioning that flavan-3-ol polyphenols, a major component in our active polar fraction, have been shown to activate a variety of cellular processes that contribute to their neuroprotective properties, such as iron chelation, survival gene and cell signaling pathway activation, and mitochondrial activity control [48–51]. As a result, these polyphenolics are anticipated to be among the most important metabolites involved in dox-induced toxicity neuroprotection.

Point 2: Besides, some sentences are unclear to me, e.g.:

„The results of the in-vivo study were corroborated with a UPLC-ESI-MS/MS profiling that revealed the accumulation of a high percentage of the plant polyphenolics.” What does it mean „accumulation”?

Response 2: The comment is considered and the previous statement has been paraphrased in the manuscript as follow:

"The results of the in-vivo study were corroborated with a UPLC-ESI-MS/MS profiling that revealed the presence of a high percentage of the plant polyphenolics with known neuroprotective properties.”

The crystal structures of TNCE (PDB ID: 2FV5)”…. What does it mean „TNCE”?

Response 2: Sorry for the typographic error; the correct is TACE, which is the abbreviation for TNF-activating converting enzyme (TACE); this was also revised in the manuscript.

Point 3: „The procyanidin B6 showed the highest affinity score with −21.9152 (RMSD = 1.25).” The interaction of the compound with the target enzyme (Figure 5) showed the presence of hydrogen bond formation with Arg 357, Pro 437, and ….”

And „Figure 5. 2D (A) and 3D (B) interactions complex of procyanidin C2 with TACE (PDB 212 ID: 2FV5)”.

I am not sure which compound is concerned: Procyanidin B6 or C2?

Response 3: Sorry for the typographic error of “C2” the sentence has been corrected

Figure 5. 2D (A) and 3D (B) interactions complex of procyanidin B6 with TACE (PDB ID: 2FV5)

Reviewer 2 Report

Dear authors,

I think this manuscript is interesting and well written.

The authors should present the link between the results and the molecular docking study.

How does this study support the biological activity of bioactive compounds?

You should show in the paper what the applicability of this study is.

Author Response

Response to Reviewer 2

First of all the authors would like to thank the reviewer for their excellent comments that increased the significance and quality of our manuscript. We addressed all comments as outlined below and all the required adjustments were considered in the manuscript.

Point 1: The authors should present the link between the results and the molecular docking study.

Response 1: We thank the reviewer for the comment, the discussion was expanded and added to the manuscript as follows:

It should be noted that TNF-α serum level, a pro-inflammatory mediator that contributed to neurotoxicity, depends on the rate of its synthesis as well as on its shedding from the cell surface (a mechanism mainly regulated by TACE). Molecular docking is frequently used to rationalize the inhibitry potential of compounds against selected target proteins and to characterize their behaviour in the binding site in order to predict the affinity and activity [52,53]. Altogether, to corroborate the reported in-vivo activity, in silico inhibitory activity experiment of variously identified 36 phytoconstituents against TACE was performed. Procyanidins are among the highest-scoring compounds as inhibitors for TACE in which the phenolic hydroxyl groups and aromatic rings impart in the interaction with the active site residues. These results, although few compounds were studied, put focus on structural characteristics that best contribute to the activity of such drugs as TACE inhibitors. The results of simulation studies are consistent with the reported activity of procyanidins, where a procyanidin enriched extract revealed a reduction in TNF-α levels and suppressed the LPS-stimulated inflammatory response in the RAW264.7 cells [54]. Further, this will add to our understanding of the flavonoid structural requirements and the pharmacophores concerned in the receptor-protein complex. Our results, in agreement with those previously detected for the same and/or analogous compounds by experimental and molecular docking studies [4,55], suggest that polyphenolic compounds of the C. macrocarpa leaves polar fraction with elevated pose score could be used as a scaffold for developing new inhibitors for enzymes related to neurotoxicity (TACE). Isolation of the predicted compounds that showed potential affinity toward TACE as well as the experimental evaluation of their activities will be conducted in future studies.

Point 2: How does this study support the biological activity of bioactive compounds?

Response 2:

Natural polyphenols have been shown to help with neurodegenerative illnesses through a variety of neuroprotective pathways. This study emphasizes the evidence that antioxidant activity, mainly inhibition of reactive oxygen species generation, reduction of neuroinflammation via attenuation of the release of cytokines and downregulation of the pro-inflammatory transcription factors are responsible for the neuroprotective actions of different natural polyphenols. Exploiting the antioxidant and anti-inflammatory characteristics of the plant rich with naturally occurring polyphenolic compounds could lead to the discovery of new therapeutic medicines to combat chemotherapeutic-induced neurotoxicity.

Point 3: You should show in the paper what the applicability of this study is.

Response 3: This point is considered into an account in conclusion section by adding:

Further studies to screen ligands that support the clinical phase (pharmacokinetics) are warranted. Additionally, dose determination, safety, and efficacy studies will be performed as an initial step for the clinical trial of C. macrocarpa leaves.

End of the comments
